# FHE-Coder: Benchmarking Secure Agentic Code Generation for Fully Homomorphic Encryption

**Mayank Kumar, Jiaqi Xue, Mengxin Zheng & Qian Lou**
Department of Computer Science
University of Central Florida
{mayank.kumar,jiaqi.xue,mengxin.zheng,qian.lou}@ucf.edu

## Abstract

Fully Homomorphic Encryption (FHE) is a foundational technology for confidential computing, yet its practical adoption remains limited by the need for specialized cryptographic expertise and error-prone parameter configuration. To lower this barrier, we investigate whether Large Language Model (LLM) agents can reliably generate secure FHE code from natural-language specifications. We present FHE-Coder, a three-phase agentic framework that addresses the key failure modes of FHE code generation: semantic ambiguity, API misuse, and cryptographic insecurity. The framework integrates (1) a Prompt Formalizer that structures user intent and enforces secure parameterization, (2) a specialized retrieval-augmented generation (RAG) module that supplies scheme-specific API and documentation knowledge, and (3) an automated Security Verifier that performs iterative validation and feedback to detect and correct cryptographic flaws. We evaluate FHE-Coder across four leading LLMs on a benchmark of ten FHE programming tasks spanning increasing functional and security complexity. While baseline agents frequently produce code that compiles and passes functional tests, they often violate security constraints or misuse cryptographic parameters. In contrast, FHE-Coder consistently generates solutions that are compilable, functionally correct, and verifiably secure across schemes including TFHE and CKKS. Our work establishes a systematic methodology and benchmark for agentic FHE code generation, providing a practical step toward democratizing secure computation without compromising cryptographic guarantees. Project page: https://fhe-coder.github.io

## 1 Introduction

Fully Homomorphic Encryption (FHE) (Gentry, 2009; Lou & Jiang, 2019; Zhang et al., 2024; Lou et al., 2021) enables computation directly over encrypted data, eliminating the need for decryption during processing (Brakerski et al., 2014; Lou & Jiang, 2021; Brakerski, 2012; Zhang et al., 2023; Fan & Vercauteren, 2012; Cheon et al., 2017b; Chillotti et al., 2020; Xue et al., 2022; Zhang et al., 2025; Lou et al., 2019a; Zheng et al., 2023). As a foundational primitive for privacy-preserving computation, it supports encrypted machine learning (Gilad-Bachrach et al., 2016; Lou et al., 2019b; Santriaji et al., 2024), secure multi-party computation (Jin et al., 2023), private blockchains (Madathil & Scafuro, 2023), and medical diagnostics (Raisaro et al., 2018).

Among modern schemes, TFHE Chillotti et al. (2020) and CKKS Cheon et al. (2017a) are widely regarded as the most industry-relevant FHE frameworks, with deployments across major technology platforms. Apple has integrated FHE-based private contact discovery and private email retrieval since iOS 18 (Apple, 2024), benefiting over 1.5 billion iPhone users as of 2025. Microsoft has also deployed FHE in its Browser Password Monitor (Research, 2020). Furthermore, Zama's FHE-based ecosystem enables encrypted machine learning (Concrete ML (Zama, 2024)), private smart contracts (fhEVM (Z1Labs, 2024)), and blockchain infrastructures (Madathil & Scafuro, 2023).

Despite this momentum, widespread adoption remains hindered by a steep learning curve. Unlike general-purpose programming, developing secure FHE applications requires precise coordination of tightly coupled cryptographic components. Developers must ensure strict parameter compatibility to satisfy Learning With Errors (LWE) security bounds while carefully managing noise growth across homomorphic operations. Even minor misconfigurations can render programs insecure or non-functional. This complexity is reflected in community benchmarks such as the 2025 FHERMA competitions (FHERMA, 2025a;b), where implementing standard activation function ($ReLU$) and convolution inference remains challenging due to noise-sensitive constraints. This expertise gap raises a fundamental question: *can LLM agents translate natural language into secure FHE code?*

Recent advancements in Large Language Models (LLMs) (Jiang et al., 2024) have demonstrated strong capabilities in natural language understanding and code generation (Mastropaolo et al., 2023; Nijkamp et al., 2022). These models can synthesize executable programs from high-level descriptions and assist developers in debugging and implementation. This progress raises the possibility of leveraging LLM agents to automate FHE programming tasks, including secure parameter configuration and correct API usage. Reliable automation would lower expertise barriers and democratize access to confidential computing.

However, naive application of general-purpose code generation agents to FHE often fails. As illustrated in Fig. 1a, agents frequently generate plaintext implementations instead of homomorphic programs (Fig. 1b). This failure arises from several domain-specific challenges. First, models trained on general-purpose code lack awareness of FHE-specific program structure and cryptographic constraints. They frequently hallucinate APIs or misuse existing functions, violating homomorphic properties. Second, parameter configuration requires scheme-specific reasoning about security levels and noise growth—capabilities absent from standard prompting pipelines. Finally, conventional evaluation metrics such as Pass@k (Chen et al., 2021) measure only functional correctness, not cryptographic security. A program may pass such tests while operating on plaintext data, entirely defeating its privacy-preserving objective. Thus, functional correctness is a flawed proxy for secure code generation.

```
No encryption of Inputs
int result = a & b;
```

**(a)** Plaintext program.

```
Parameter Setup & Key Generation
const int minimum_lambda = 110;
TFheGate...ParameterSet* params = ...;
TFheGate...SecretKeySet* key = ...;

Encryption of Input
bootsSymEncrypt(&ctx_a[i], ...);

Homomorphic Operation
bootsAND(&result_ctx[i],...);

Decryption of Output
bootsSymDecrypt(&result_ctx[i], ...);
```

**(b)** TFHE program.

**Figure 1:** Illustrating the differences between a plaintext program and a TFHE-based FHE program.

To address these limitations, we introduce a novel agentic workflow and evaluation framework for secure FHE code generation, illustrated in Fig. 2. Our workflow consists of three tightly integrated components designed to mitigate the identified failure modes. First, the **FHE Prompt Formalizer** (Fig. 3a) translates user intent into a structured specification with securely derived cryptographic parameters. Second, the **FHE API RAG Retriever** (Fig. 3b) supplies scheme-specific documentation and expert-curated usage examples to prevent API hallucination and misuse. Third, the **FHE Security Verifier** (Fig. 4) introduces an automated feedback loop that enforces security invariants and detects plaintext leakage or parameter inconsistencies. Together, these components transform unconstrained code generation into a security-aware, iterative synthesis process.

We summarize our contributions as follows:

- We propose **FHE-Coder**, an agentic framework that enables reliable and security-aware FHE code generation using LLM agents. It introduces three core components: an **FHE Prompt Formalizer** that derives parameters grounded in LWE-based security analysis, an **Expert-Enriched RAG Retriever** that bridges gaps in cryptographic API knowledge, and an **FHE Security Verifier** that enforces privacy invariants through automated validation.
- We show traditional functional metrics are insufficient for private computing and introduce a **rigorous security evaluation methodology**, featuring a new **Pass@1 (func_sec)** metric and a multi-stage pipeline ensuring functional and cryptographic correctness.

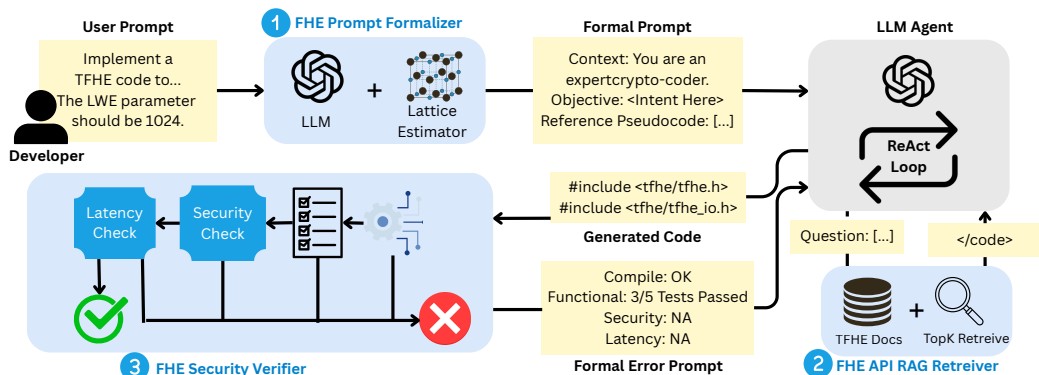

**Figure 2:** Overview of our secure FHE code generation workflow. Our key components (highlighted with stars) are: (1) the FHE Prompt Formalizer, which enriches a developer's prompt with securely derived parameters using a Lattice Estimator; (2) the FHE API RAG Retriever, which provides expert-annotated API usage examples; and (3) the FHE Security Verifier, which establishes an automated feedback loop for security and correctness. Maximum iterations are set to 10.

- We establish a **comprehensive benchmark for agentic FHE generation** across TFHE and CKKS. We show that while baselines fail security-critical tasks, our framework excels on complex architectures (e.g., Transformers), offering a blueprint for automated secure software engineering.

## 2 BACKGROUND

### 2.1 FULLY HOMOMORPHIC ENCRYPTION

Fully Homomorphic Encryption (FHE) enables arbitrary computation over encrypted data without requiring decryption, providing strong confidentiality guarantees grounded in hard lattice problems such as Learning With Errors (LWE). Modern FHE schemes can be broadly categorized into boolean-gate-based systems (e.g., TFHE) and arithmetic-circuit-based systems (e.g., BGV (Brakerski et al., 2014) and CKKS (Cheon et al., 2017b)).

TFHE (Fully Homomorphic Encryption over the Torus) (Jiang et al., 2022) operates over boolean circuits and supports logical gates (NOT, AND, OR) with highly efficient Programmable Bootstrapping (PBS), enabling evaluation of non-linear functions while refreshing ciphertext noise. In contrast, arithmetic schemes like BGV and CKKS operate over polynomial rings, optimizing integer or real-number computation for encrypted ML workloads.

Despite their structural differences, these schemes share common cryptographic constraints. Developers must carefully manage noise growth across homomorphic operations, ensure strict compatibility among interdependent security parameters (e.g., security level $\lambda$, lattice dimension, modulus size), and avoid silent decryption failures caused by parameter misconfiguration. In CKKS, additional challenges arise from approximation error control and rescaling strategies, while TFHE requires precise bootstrapping configuration and gate-level orchestration.

Consequently, although modern FHE schemes provide powerful functionality, implementing secure and efficient FHE applications requires specialized cryptographic expertise. This complexity hinders widespread adoption across both boolean and arithmetic FHE.

### 2.2 LLM AGENTS FOR CODE GENERATION

Code generation has emerged as a core application of large language models, with systems such as CodeGen (Nijkamp et al., 2022), CodeX (Chen et al., 2021), and CodeT5 (Wang et al., 2021) demonstrating strong performance in mainstream programming languages due to the availability of large-scale training corpora. However, generating secure code for niche crypto libraries is harder due to sparse documentation, domain-specific APIs, and mathematical constraints.

Recent advances extend beyond standalone LLMs toward *LLM agents*, which integrate planning, retrieval, and tool-use capabilities into iterative reasoning pipelines. In adjacent domains such as High-Level Synthesis (HLS) and Register Transfer Level (RTL) hardware design (Thakur et al., 2023; Liao et al., 2024; Xiong et al., 2024), LLM-based systems have demonstrated the ability to

reason about logical operations (e.g., AND/OR gates) and structured transformations. his suggests LLM agents possess reasoning that aligns with circuit-oriented FHE schemes.

However, unlike standard program synthesis tasks, FHE code generation imposes additional security-critical constraints: correctness depends not only on syntactic validity and functional behavior, but also on cryptographic soundness, parameter security margins, and ciphertext-only computation guarantees. Traditional prompting cannot enforce these invariants. This motivates using structured, security-aware LLM agents with parameter derivation, API grounding, and verification to reliably generate FHE programs for schemes like TFHE and CKKS.

## 3 OUR METHOD: FHE-CODER

Our Agentic workflow 2 has three main components: **FHE Prompt Formalizer**, **FHE API RAG Retriever** and **FHE Security Verifier**. The Agent uses ReAct (Yao et al., 2023) prompting.

### 3.1 FHE PROMPT FORMALIZER

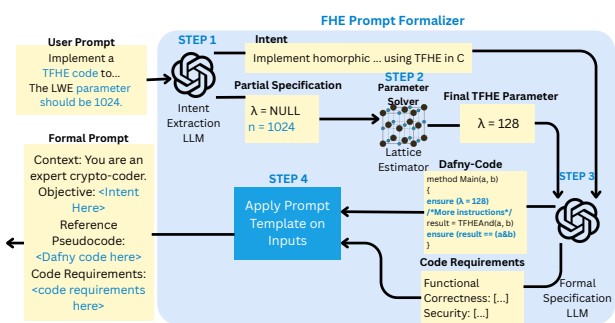

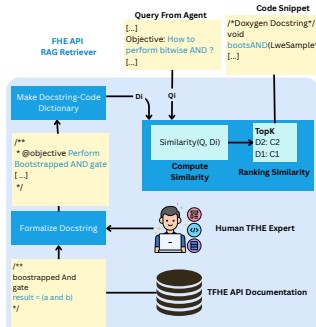

**(a)** LLMs and a Lattice Estimator transform a developer's prompt into a secure and structured set of instructions. The process extracts intent and solves for cryptographic parameters.

**(b)** An offline, human-in-the-loop process maps expert-enriched docstrings to TFHE code snippets.

**Figure 3:** Overview of the instruction and generation phase. (a) The online instruction generation and parameter estimation. (b) The offline code mapping process.

The FHE Prompt Formalizer, illustrated in Figure 3a, addresses a critical failure mode in baseline agents: the inability to select secure cryptographic parameters through probabilistic generation. While standard LLMs often hallucinate inconsistent parameters or insecure noise budgets due to a lack of domain-specific mathematical structure, our workflow replaces this heuristic guesswork with cryptographic certainty. The process begins by extracting the user's high-level `intent`, which is then passed to a `Lattice Estimator` (Albrecht et al., 2015) to mathematically solve for the precise security parameter, $\lambda$, rather than predicting it. A second LLM utilizes this secure configuration to generate a formal specification containing `Dafny` (Leino, 2010) pseudocode. This step transforms ambiguous natural language into rigorous structural constraints; for instance, the generated `ensure` statements explicitly guide the agent to embed `assert` checks in the final C++ code, ensuring the solution adheres to valid ciphertext structures and intermediate invariants.

While our FHE-Coder framework demonstrates strong performance on atomic tasks, Figure 6 shows that direct generation degrades on complex, compositional problems that require multi-step reasoning (e.g., matrix-vector multiplication, MLP). To address this, we use a ***structured decomposition*** strategy: the agent first generates secure primitives (e.g., dot product); then, a "composer" agent receives these verified subroutines and the high-level goal to build the final program.

### 3.2 FHE API RAG RETRIEVER

The FHE API RAG Retriever, illustrated in Figure 3b , addresses the limitations of standard retrieval methods, which fail almost entirely in this domain because LLMs lack the intrinsic structure to interpret strict cryptographic APIs or respect ciphertext-only computation rules B. To bridge

the semantic gap between natural-language intent and these rigid library constraints, we construct a knowledge base using expert-enriched metadata. Specifically, we transform FHE method (e.g., TFHE) docstrings[1] into the `Doxygen` format[2], utilizing structured tags such as `@objective` to embed machine-readable semantic instructions. This enrichment enables the agent to retrieve precise, security-compliant code snippets based on cryptographic purpose rather than ambiguous keyword matching, ensuring the selected APIs adhere to necessary noise and parameter rules. Crucially, this offline preparation is a one-time, lightweight security step designed to eliminate API hallucinations, making the process fully scalable. Extending the framework to another scheme (e.g., CKKS) simply requires replacing the documentation corpus with that scheme's library documentation, rendering cross-scheme adaptation essentially plug-and-play. In our experiments, the chunking-size was set to 600, and the chunk-overlap was set to 120.

## 3.3 FHE Security Verifier

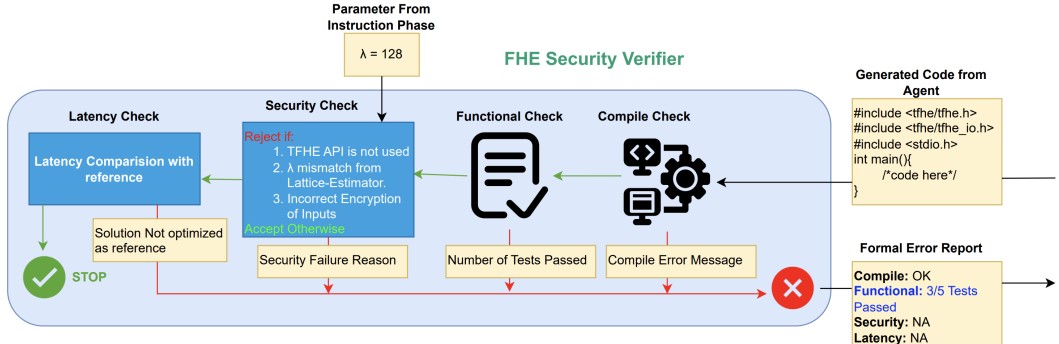

**Figure 4:** This automated pipeline validates generated code across four stages: Compile, Functional, Security, and Latency. The critical Security Check verifies correct API usage, secure parameter configuration, and proper input encryption. If any stage fails, a consolidated Formal Error Report is generated and returned to the agent for iterative correction; otherwise, the solution is accepted.

The FHE Security Verifier, detailed in Figure 4, introduces a novel, complementary validation layer that addresses the critical limitation of standard code generation metrics. While traditional Pass@k functional testing (implemented here via unit tests) ensures the code aligns with the user's operational intent, our experiments show that baseline LLMs often produce code that passes these functional tests while remaining cryptographically insecure (e.g., using plaintext operations). To bridge this gap, our pipeline subjects the agent's code to a specialized Security Check rooted in the mathematical hardness of the Learning With Errors (LWE) problem. Unlike simple heuristic checks, this module strictly enforces LWE-based parameter bounds derived from the Lattice Estimator and mandates exclusive homomorphic API usage to prevent data leakage. When a violation is detected alongside Compile, Functional, or Latency errors, the system generates a structured `Formal Error Report`, driving an automated feedback loop that forces the agent to converge on a solution that is simultaneously functionally accurate and mathematically secure.

## 4 EXPERIMENT DESIGN SECTION

### 4.1 PROBLEM DEFINITION

We introduce the LLM-Agentic FHE Generation and Evaluation Framework, illustrated in Figure 2. In this framework, each TFHE task is formulated as a natural language prompt and provided to the agent. Leveraging its reasoning capabilities, the agent may (optionally) consult external documentation through retrieval-augmented generation (RAG) before implementing the corresponding code. The generated code is subsequently evaluated by the Security Verifier, which consolidates any compilation or verification errors into structured feedback and returns it to the agent for refinement. The agent iteratively revises its solution until security checks pass or the 10-step limit is reached.

---

[1]https://tfhe.github.io/tfhe/gate-bootstrapping-api.html
[2]https://www.doxygen.nl/

## 4.2 WORKLOAD SELECTION

We describe the workloads for the code generation breifly in Table 1.

Table 1: Benchmark workloads for evaluating TFHE code generation, categorized by complexity.

| Complexity | Workload | Description |
|---|---|---|
| **Primitives** | AND | Bitwise AND between two 32-bit integers. |
| | ReLU | ReLU on a signed 32-bit integer. |
| | Adder | Addition of two 32-bit integers. |
| | Multiplier | Multiplication of two 32-bit integers. |
| **Linear Algebra** | Vector Add | Vector addition between two integer vectors of length 5. |
| | Dot Product | Inner-product of two integer vectors of length 5. |
| | Mat-Vec Mult | Multiplication between an encrypted vector and a plaintext matrix. |
| | Mat-Mat Mult | Multiplication between an encrypted matrix and a plaintext matrix. |
| **Deep Learning** | MLP | A simple 3-layer MLP with ReLU activation. |
| | CNN | A small Convolutional Neural Network with a fully connected layer. |
| | Softmax | Computes the Softmax activation function over an input vector. |
| | Attention | Single-head self-attention mechanism (Query, Key, Value). |
| | Transformer | A complete Transformer block combining attention and feed-forward layers. |

These tasks collectively evaluate LLMs' ability to synthesize both low-level cryptographic primitives and high-level machine learning components using TFHE's gate-level programming paradigm. However, in Section 5.3, we evaluate more complex workloads as well.

## 4.3 MODEL SELECTION

We select the latest LLMs to drive our agent. For open-source LLMs, we choose Qwen3-Coder-480B-A35B( (Yang et al., 2025)) (QWE) and Deepseek-V3.1( (Liu et al., 2024))(DSK). For closed-source LLMs, we select Gemini-2.5-Pro( (Comanici et al., 2025))(GEM) and GPT-5( (OpenAI, 2025))(GPT). For all studied LLMs, we set the temperature to 0.5. Furthermore, there is no heavy local resource consumption for generation, as models are accessed via the OpenRouter API.

Note that, to mitigate issues stemming from the randomness of model generation, the experimental results presented in this paper are obtained by conducting five repeated experiments and averaging the results. For RAG, we used OpenAI's `text-embedding-3-small`[3].

## 4.4 METRICS

In our framework, we employ three key metrics to judge the quality of the generated codes. Following prior works on code generation, we use **1.** *Pass@k(func)* to denote the fraction of generated codes that pass the unit tests. We present our novel metric **2.** *Pass@k(func_sec)*, which denotes the fraction of generated codes that are both functionally correct and cryptographically secure. A program is considered **secure** only if it passes an automated analysis verifying: **(i)** exclusive use of TFHE APIs to prevent plaintext data leakage, **(ii)** correct configuration of cryptographic parameters against secure values from the Lattice Estimator, and **(iii)** proper encryption of all inputs before their use. A failure in any of these checks renders the code insecure. Our third metric is **3.** *Latency*, compared to expert-written reference codes. This is computed as the ratio of execution times of the generated solution and the expert-written solution.

## 4.5 BASELINES

Our first baseline denotes a **regular code generation** workflow. Concretely, given the same natural-language task description, the agent directly generates TFHE code in a single shot (or with its default self-revision), *without* (i) the Prompt Formalizer (no enforced program skeleton or parameter

---

[3]https://platform.openai.com/docs/models/text-embedding-3-small

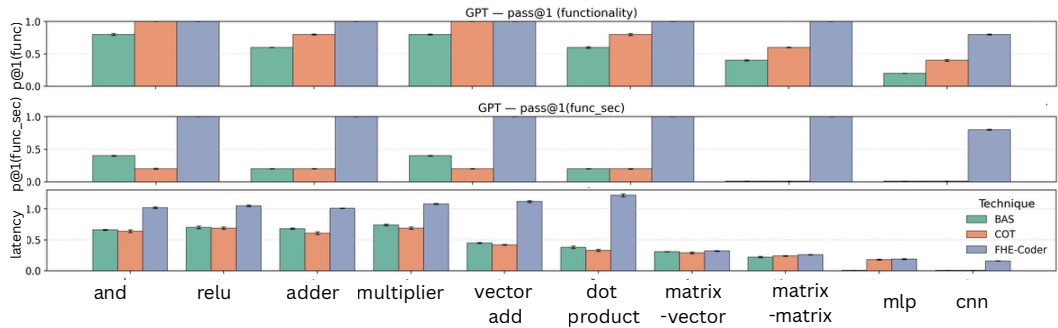

**Figure 5:** Performance of GPT-5 across representative tasks. A comparison of our framework (FHE-Coder) against Baseline (BAS) and Chain-of-Thought (COT) techniques. FHE-Coder consistently produces code that is both functionally correct and verifiably secure.

derivation), (ii) the API RAG Retriever (no documentation-grounded API lookup), and (iii) the Security/Latency verification loop (no iterative feedback from compilation, functional tests, or security checks as in Fig. 2).We abbreviate it as BAS. Our second baseline is **Zero-shot Chain-of-Thought** agent, which builds upon the regular workflow by appending a step-by-step worked example of correct TFHE code generation. We abbreviate it as COT.

## 5 EVALUATION RESULTS

This section presents a comprehensive empirical analysis of our proposed agentic framework (denoted as FHE-Coder) in comparison to a regular code generation workflow (BAS) and a Zero-shot Chain-of-Thought agent (COT). We first evaluated four leading LLMs on a benchmark of ten TFHE programming tasks with varying complexity, and then extended the scheme to CKKS.

### 5.1 IN-DEPTH ANALYSIS ON A STATE-OF-THE-ART MODEL

We first conduct a detailed analysis using the state-of-the-art GPT-5 model to illustrate the core performance differences between our framework and the baselines across all tasks. The results are presented in Fig. 5.

**Functional Correctness:** As shown in the top chart of Figure 5, the baseline methods (BAS and COT) demonstrate partial success on tasks with low complexity, such as `and` and `relu`. We defer discussion of tougher, compositional tasks to Section 5.3.

**Security:** A critical distinction between our framework and the baselines is revealed in this evaluation. Both the BAS and COT methods yield a `pass@1(func_sec)` approaching zero for all tasks evaluated. This finding indicates that they consistently fail to produce secure code, often generating plaintext implementations that, while sometimes functionally correct, do not adhere to the required cryptographic protocols. Conversely, the FHE-Coder framework achieves near-perfect scores across the entire benchmark. This outcome underscores the necessity of a guided, multi-phase process—encompassing the proposed prompt formalization, accurate API retrieval, and security verification—to meet the specific requirements of secure code generation.

**Latency:** The performance trade-offs are detailed in the bottom chart of Figure 5. While the FHE-Coder framework naturally incurs higher latency due to its iterative feedback loop, the specific computational overhead introduced by our verification logic is minimal. The **Compile** and **Security Checks** rely on lightweight static analysis and complete in the order of milliseconds, while the **Latency Check** adds negligible overhead. The majority of the time is consumed by the **Functional Check**, which ranges from seconds (for simple gates) to minutes (for complex circuits like Matrix Multiplication), but this duration is intrinsic to the execution of the generated FHE code itself rather than the framework's processing. This overhead is a deliberate design choice, representing a practical trade-off for significant improvements in security. The absolute execution runtimes of generated codes of the representative workloads are mentioned in Table 2.

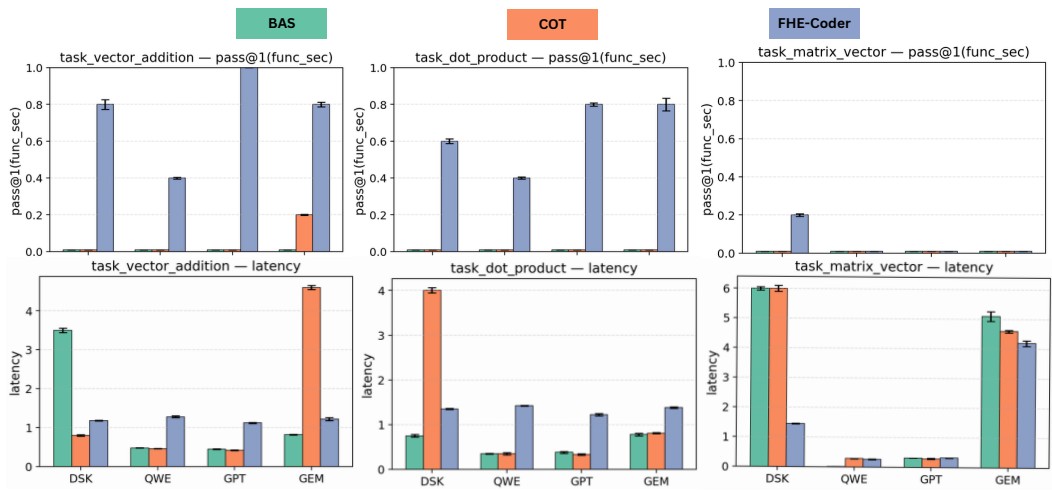

**Figure 6:** Comparison of FHE-Coder against baselines (BAS, COT) across four LLMs. Baselines universally fail security checks, whereas FHE-Coder consistently ensures security, showing model-agnostic benefits. Note: Latency measures the relative runtime overhead of the generated code.

## 5.2 GENERALIZABILITY ACROSS DIVERSE LLMs

To ensure our findings are not model-specific, we assessed the generalizability of our framework by applying it to four different LLMs. Fig. 6 presents a comparative analysis on three representative tasks. The results confirm that the performance patterns persist across all models. The security deficiencies of the baseline methods are model-agnostic; both BAS and COT fail to generate secure code regardless of the LLM used. In contrast, the FHE-Coder framework is the only approach that enables the models to consistently produce functional and secure outputs. While the overall performance ceiling is influenced by the base model's intrinsic capabilities—with GPT-5 and Gemini-2.5-pro generally outperforming Deepseek-V3.1 and Qwen3-Coder —our framework provides a consistent and essential security impoves for all tested models.

## 5.3 SOLVING COMPLEX TASKS WITH STRUCTURED DECOMPOSITION

As detailed in Figure 7, our structured decomposition approach effectively solves tougher compositional tasks (e.g., matrix-vector multiplication and CNN) where the direct method fails. By integrating structured decomposition (With SD), the agent achieves strong pass@1(func_sec) scores across most tasks—reaching 0.70 for matrix-vector multiplication and

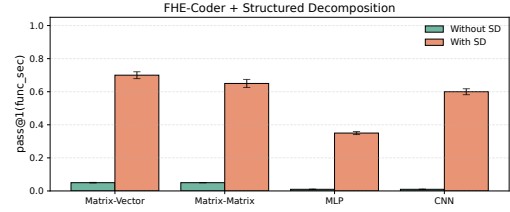

**Figure 7:** Performance: FHE-Coder + structured decomposition.

0.60 for CNNs—whereas direct generation (Without SD) yields near-zero success. The increased compositional complexity of the MLP task lowers the combined pass rate to approximately 0.35, suggesting a link between logical circuit depth and the agent's ability to simultaneously maintain functionality and security protocols. **Overall, this hierarchical strategy of composing verified sub-tasks extends the agent's capabilities to previously unsolvable problems.**

**Scaling to Complex Non-Linear TFHE Architectures.** We further test this decomposition strategy on complex non-linear TFHE circuits (Softmax, Attention, and Transformer). Figure 10 shows that baselines (BAS, COT) consistently fail to produce valid and secure code, yielding a pass@1(func_sec) of approximately 0.0. In contrast, FHE-Coder combined with structured decomposition achieves perfect pass@1(func_sec) scores on Softmax and Attention for the GPT model. However, the reduced pass rate on Transformers (e.g., $\approx 0.4$ for GPT) and DSK's failure

on Softmax indicate that while the framework successfully scales to deeper circuits, overall performance remains partially constrained by the base LLM's intrinsic reasoning capabilities.

## 5.4 GENERALIZATION TO CKKS SCHEME

Figure 9 shows that our workflow generalizes beyond TFHE to CKKS on non-linear workloads (Softmax, Attention, Transformer). Baselines (BAS, COT) yield near-zero `pass@1(func_sec)` scores across all tasks. In contrast, FHE-Coder—using a CKKS parameter estimator and documentation achieves perfect `pass@1(func_sec)` results on Attention for both models, and on Softmax for GPT. However, DSK's failure on Softmax and both models' moderate Transformer performance ($\approx 0.4$) indicate that while the framework adapts to new cryptographic schemes, overall success remains constrained by the underlying LLM's intrinsic limits.

## 5.5 ABLATION STUDY

To quantitatively assess the contribution of each component within our framework, we conduct a component analysis by systematically integrating each of our three key modules: the FHE Prompt Formalizer (FP), the FHE API RAG Retriever (RAG), and the FHE Security Verifier. The results for the GPT-5 model on the representative Vector Addition task, shown in Figure 8, reveal a clear cumulative performance hierarchy.

The study establishes that the Baseline (BAS) agent, lacking our modules, is unable to produce valid and secure TFHE code, yielding a `pass@1(func_sec)` score of approximately 0.0. Integrating the FHE Prompt Formalizer (+ FP) provides the initial structural guidance and secure parameter derivation necessary to elevate the `pass@1(func_sec)` score to 0.6. Subsequently adding the FHE API RAG Retriever (+ FP + RAG) supplies the agent with precise cryptographic documentation, further improving the score to 0.8 by mitigating API hallucinations. Finally, the full FHE-Coder framework, which incorporates the automated feedback loop of the FHE Security Verifier, is the only configuration to achieve a perfect 1.0 `pass@1(func_sec)` score. This confirms that all three components are essential and complementary, working in concert to enforce both functional correctness and strict cryptographic security.

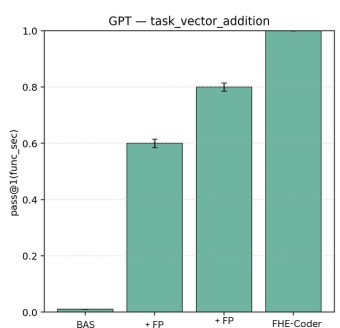

**Figure 8: Ablation Study on Vector Addition.** Component Analysis on Vector Addition. Progressively adding modules (+ FP, + RAG) improves the baseline `pass@1(func_sec)` score, but the full FHE-Coder framework is required to achieve a perfect 1.0.

**Table 2:** Illustrative absolutte execution Times for FHE workloads.

| Task (Complexity) | Average Execution Time |
|---|---|
| **AND** (Elementary Gate Logic) | 10.45 milliseconds |
| **ReLU** (Simple Bitwise Circuit) | 11.50 milliseconds |
| **MatMul** (Complex Compositional Circuit) | 2.54 minutes |

## 6 DISCUSSION

Our work demonstrates that automating the generation of secure FHE code requires moving beyond simple prompt-to-code workflows, which we show consistently fail in security-critical domains. The core finding is that while baseline agents may achieve functional correctness, they lack the intrinsic mathematical reasoning to select secure parameters or adhere to strict ciphertext-only constraints, resulting in a near-zero security pass rate. This highlights a fundamental misalignment: traditional metrics like *Pass@k(func)* are misleading proxies for success in cryptographic programming. Our framework addresses this by elevating verifiable security to a primary objective, grounding the generation process in the mathematical hardness of the Learning With Errors (LWE) problem via the Lattice Estimator, rather than relying on probabilistic LLM heuristics.

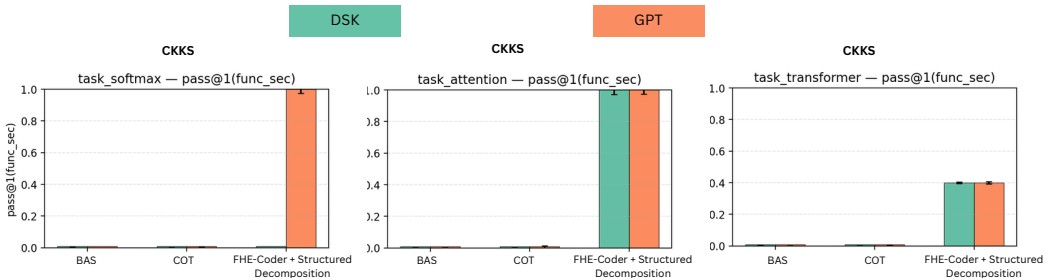

**Figure 9: Generalization to CKKS.** FHE-Coder + Structured Decomposition significantly outperforms failing baselines on non-linear architectures. While GPT achieves high `pass@1(func_sec)` across all tasks, DSK's failure on Softmax indicates that the framework's effectiveness remains partially constrained by the base LLM's intrinsic capabilities.

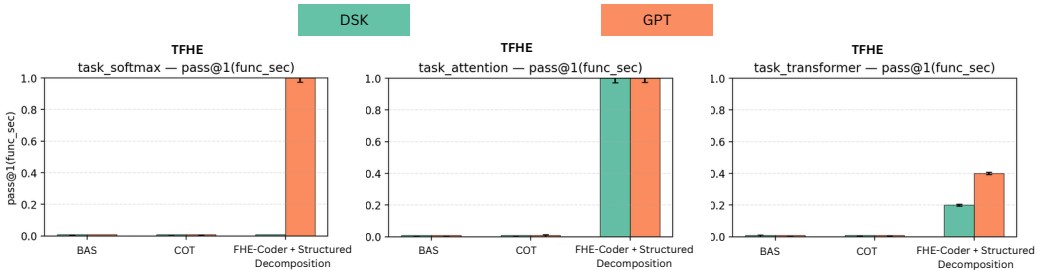

**Figure 10: Complex TFHE Tasks.** FHE-Coder + Structured Decomposition significantly outperforms failing baselines (BAS, COT) on non-linear circuits. While it enables perfect `pass@1(func_sec)` scores for GPT on Softmax and Attention, DSK's failure on Softmax and both models' lower Transformer scores indicate that base LLM capabilities can still constrain performance on deeper circuits.

Crucially, our extended evaluation confirms that this agentic architecture serves as a generalizable blueprint for confidential computing, not just a TFHE-specific solution. The modular design of our three components—Prompt Formalizer, RAG Retriever, and Security Verifier—enables "plug-and-play" adaptation to other schemes. As evidenced by our results with CKKS, we achieved high security scores by simply swapping the underlying parameter solver and documentation corpus, proving that the workflow's effectiveness is scheme-agnostic. Furthermore, our experiments with Structured Decomposition reveal that the framework scales to state-of-the-art complexity. While direct generation struggles with deep circuits, agents excel as "composers" when provided with verified primitives, successfully implementing complex non-linear architectures like Attention mechanisms and Transformers. FHE-Coder demonstrates that with structural knowledge and security feedback, LLMs can evolve from unreliable assistants into verifiable partners for secure software engineering.

## 7 CONCLUSION

Fully Homomorphic Encryption has emerged as a cornerstone of modern confidential computing, enabling unique programmable bootstrapping capabilities that drive widespread industry adoption; however, utilizing this powerful tool is severely limited by a steep learning curve. In this work, we introduced FHE-CODER, a structured agentic framework that bridges this gap by integrating prompt formalization, retrieval-augmented generation, and a critical security verification loop. Our findings reveal that while standard agents consistently fail due to a lack of domain constraints, our framework succeeds by strictly grounding the generation process in the mathematical hardness of the Learning With Errors (LWE) problem. Beyond establishing the first robust benchmark for TFHE, our extended evaluation demonstrates significant generalization and scalability: we proved that the framework's modular design serves as a scheme-agnostic blueprint, successfully adapting to the CKKS scheme, and that a structured decomposition strategy enables the synthesis of state-of-the-art non-linear architectures, such as Attention mechanisms and Transformers. By enabling developers to reliably generate code that is simultaneously functional and verifiably secure, this work takes a definitive step toward democratizing access to these advanced privacy-preserving technologies.

## 8 ETHICS STATEMENT

Large language models (LLMs) were used to check grammer and polish writing.

## 9 REPRODUCIBILITY STATEMENT

In order to ensure reproducibility, we fixed the seed while inferencing the models. As noted in Section 4.3, the results of the experiments shown are aggregated after 5 independent runs with different seed values. Source code & leaderboard: `https://fhe-coder.github.io/`

## 10 ACKNOWLEDGEMENTS

This material is based upon work supported by the National Science Foundation under Grant Nos. CCF-2523407 and CNS-2413232. Any opinions, findings, and conclusions or recommendations expressed in this material are those of the author(s) and do not necessarily reflect the views of the National Science Foundation.

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

# A APPENDIX

## A FULL CODE EXAMPLE: BITWISE RELU

Below is a full working example for a bitwise ReLU operation. Listing 1 provides a complete, runnable C program demonstrating the plaintext bitwise logic, which reads an integer from standard input. Listing 2 shows the full, equivalent implementation using the TFHE library.

```c
1  #include <stdio.h>
2  #include <stdint.h>
3
4  // Plaintext bitwise ReLU for a 32-bit signed integer.
5  int32_t relu(int32_t input) {
6      int32_t mask = input >> 31;
7      int32_t output = input & ~mask;
8      return output;
9  }
10
11 int main() {
12     int32_t input_val;
13
14     scanf("%d", &input_val);
15
16     printf("%d\n", relu(input_val));
17
18     return 0;
19 }
```

Listing 1: Complete, runnable plaintext C code for bitwise ReLU.

```c
1  #include <tfhe/tfhe.h>
2  #include <tfhe/tfhe_io.h>
3  #include <stdio.h>
4  #include <assert.h>
5
6  // The plaintext relu function would be included here
7  int32_t relu(int32_t input);
8
9  int main() {
10     // 1. Generate a keyset
11     const int minimum_lambda = 110;
12     TFheGateBootstrappingParameterSet* params =
       new_default_gate_bootstrapping_parameters(minimum_lambda);
13
14     // Generate a deterministic key for reproducibility
15     uint32_t seed[] = { 314, 1592, 657 };
16     tfhe_random_generator_setSeed(seed, 3);
17     TFheGateBootstrappingSecretKeySet* key =
       new_random_gate_bootstrapping_secret_keyset(params);
18
19     int32_t plaintext1;
20     scanf("%d", &plaintext1);
21
22     // 2. Encrypt the 32-bit signed integer bit by bit
23     LweSample* ciphertext1 = new_gate_bootstrapping_ciphertext_array(32,
       params);
24     for (int i = 0; i < 32; i++) {
25         bootsSymEncrypt(&ciphertext1[i], (plaintext1 >> i) & 1, key);
26     }
27
28     // 3. Homomorphically compute ReLU: result = input & ~mask
29     LweSample* result = new_gate_bootstrapping_ciphertext_array(32,
       params);
30     LweSample* mask = new_gate_bootstrapping_ciphertext_array(32, params)
       ;
31
32     // 3a. Create a 32-bit mask from the encrypted sign bit (bit 31)
33     for (int i=0; i<32; i++){
34         bootsCOPY(&mask[i], &ciphertext1[31], &key->cloud);
35     }
36
37     // 3b. Invert the mask homomorphically
38     for (int i=0; i<32; i++){
```

```
39          bootsNOT(&mask[i], &mask[i], &key->cloud);
40      }
41
42      // 3c. Compute the final result: input & ~mask
43      for (int i = 0; i < 32; i++){
44          bootsAND(&result[i], &ciphertext1[i], &mask[i], &key->cloud);
45      }
46
47      // 4. Decrypt the result for verification
48      int32_t final_result = 0;
49      for (int i = 0; i < 32; i++) {
50          int bit = bootsSymDecrypt(&result[i], key);
51          final_result |= (bit << i);
52      }
53      printf("%d\n", final_result);
54
55      // Verify against the plaintext function
56      assert(final_result == relu(plaintext1));
57
58      // Cleanup
59      delete_gate_bootstrapping_ciphertext_array(32, mask);
60      delete_gate_bootstrapping_ciphertext_array(32, result);
61      delete_gate_bootstrapping_ciphertext_array(32, ciphertext1);
62      delete_gate_bootstrapping_secret_keyset(key);
63      delete_gate_bootstrapping_parameters(params);
64
65      return 0;
66 }
```

**Listing 2:** Full secure TFHE implementation for bitwise ReLU.

# B    USING VANILLA RAG

**Table 3:** Results for Using Vanilla RAG with Documentation (without preprocessing with expert summary). Standard RAG fails to provide robust security or functionality.

| Workload | pass@1 (func.) | pass@1 (sec.) |
|----------|----------------|---------------|
| ReLU     | $0.40 \pm 0.02$ | $0.20 \pm 0.01$ |
| MatMul   | $0.20 \pm 0.01$ | $0.00 \pm 0.00$ |

# C    RESULTS ON ADDITIONAL TASKS (AND SCHEMES)

**Table 4:** Comparison of functional correctness and security Pass@1 rates for TFHE workloads using GPT-5 across different prompting strategies (BAS, COT, and FRS).

| Workload | BAS p@1 (func.) | BAS p@1 (sec.) | COT p@1 (func.) | COT p@1 (sec.) | FHE-Coder p@1 (func.) | FHE-Coder p@1 (sec.) |
|----------|-----------------|----------------|-----------------|----------------|-----------------------|----------------------|
| MatMul | $0.00 \pm 0.00$ | $0.00 \pm 0.00$ | $0.00 \pm 0.00$ | $0.00 \pm 0.00$ | $0.80 \pm 0.05$ | $1.00 \pm 0.00$ |
| Softmax | $0.80 \pm 0.05$ | $0.00 \pm 0.00$ | $0.60 \pm 0.02$ | $0.00 \pm 0.00$ | $1.00 \pm 0.00$ | $1.00 \pm 0.00$ |
| Attention | $0.00 \pm 0.00$ | $0.00 \pm 0.00$ | $0.20 \pm 0.01$ | $0.00 \pm 0.00$ | $0.80 \pm 0.02$ | $0.80 \pm 0.02$ |
| Transformer | $0.00 \pm 0.00$ | $0.00 \pm 0.00$ | $0.20 \pm 0.01$ | $0.00 \pm 0.00$ | $0.40 \pm 0.01$ | $0.80 \pm 0.02$ |

**Table 5:** Comparison of functional correctness and security Pass@1 rates for CKKS workloads using GPT-5 across different prompting strategies (BAS, COT, and FHE-Coder).

| Workload | BAS p@1 (func.) | BAS p@1 (sec.) | COT p@1 (func.) | COT p@1 (sec.) | FHE-Coder p@1 (func.) | FHE-Coder p@1 (sec.) |
|----------|-----------------|----------------|-----------------|----------------|-----------------------|----------------------|
| MatMul | $1.00 \pm 0.00$ | $0.00 \pm 0.00$ | $0.80 \pm 0.01$ | $0.00 \pm 0.00$ | $0.80 \pm 0.01$ | $0.60 \pm 0.02$ |
| Softmax | $1.00 \pm 0.00$ | $0.00 \pm 0.00$ | $0.80 \pm 0.03$ | $0.00 \pm 0.00$ | $0.80 \pm 0.02$ | $0.60 \pm 0.02$ |
| Attention | $0.40 \pm 0.02$ | $0.00 \pm 0.00$ | $0.40 \pm 0.01$ | $0.00 \pm 0.00$ | $1.00 \pm 0.00$ | $1.00 \pm 0.00$ |
| Transformer | $0.00 \pm 0.00$ | $0.00 \pm 0.00$ | $0.20 \pm 0.01$ | $0.00 \pm 0.00$ | $0.40 \pm 0.01$ | $0.60 \pm 0.03$ |

## D  SAMPLE FORMAL PROMPTS: AND TASK

To concretely illustrate the output of the FHE Prompt Formalizer (Section 3.1), we show the developer-supplied natural-language description and the corresponding structured formal prompt generated for the AND task (bitwise AND of two 32-bit integers).

**Developer input.**

```
Implement homomorphic bitwise AND between two 32-bit integers using TFHE.
The LWE parameter should be 1024.
```

**Formal prompt (output of FHE Prompt Formalizer).**

```
Context:
  You are an expert crypto-coder with deep knowledge of the TFHE C library.
  Generate only ciphertext-compliant code using the TFHE gate-bootstrapping
  API. Do not use plaintext values in any homomorphic operation.

Objective:
  Implement a bitwise AND operation on two encrypted 32-bit signed integers.
  Each integer is represented as an array of 32 LweSample ciphertexts,
  where bit i of plaintext x is stored in ciphertext[i].

Security Parameters (derived via Lattice Estimator):
  minimum_lambda = 110    (targeting lambda = 128-bit LWE security)
  Use: new_default_gate_bootstrapping_parameters(minimum_lambda)
  Use: new_random_gate_bootstrapping_secret_keyset(params)

Code Requirements:
  Functional:
    for all i in [0, 32):
      decrypt(result[i]) == decrypt(a[i]) & decrypt(b[i])
  Security:
    (i)  All inputs must be encrypted with bootsSymEncrypt before any
         gate call.
    (ii) No plaintext integers may be passed directly to gate operations.
    (iii) Cryptographic parameters must satisfy minimum_lambda >= 110.

Reference Pseudocode (Dafny formal specification):
  method BitwiseAND(a:     array<LweSample>,
                    b:     array<LweSample>,
                    key:   TFheGateBootstrappingSecretKeySet)
    returns (result: array<LweSample>)
    requires a.Length == 32 && b.Length == 32
    requires forall i :: 0 <= i < 32 ==>
               IsEncrypted(a[i], key) && IsEncrypted(b[i], key)
    ensures result.Length == 32
    ensures forall i :: 0 <= i < 32 ==>
               Decrypt(result[i], key) ==
               (Decrypt(a[i], key) & Decrypt(b[i], key))
  {
    result := new LweSample[32];
    for i := 0 to 31 {
      bootsAND(result[i], a[i], b[i], key.cloud);
      assert IsEncrypted(result[i], key);
    }
  }
```

The four fields — Context, Objective, Security Parameters, and Reference Pseudocode – are generated by the Formalizer. Note that the user's original (incorrect) LWE value of 1024 is silently overridden: the Lattice Estimator determines that `minimum_lambda=110` is the correct parameter to achieve the target security level, and this derived value appears in the formal prompt instead. The RAG Retriever then augments this prompt with relevant TFHE API snippets (e.g., `bootsAND` usage examples) before it is delivered to the ReAct agent.

## D  UNIT TEST CREATION

**Test unit construction.** For each benchmark task, a fixed set of 5 unit tests is manually written by the authors (as illustrated in the Formal Error Report in Figure 4, which shows "Functional: 3/5 Tests Passed"). Each test specifies a concrete input drawn from the task's valid input domain — for example, a pair of 32-bit signed integers for AND and Adder, an integer vector of length 5 for Dot Product and Vector Addition, or an integer matrix for Mat-Vec and Mat-Mat — executes the generated FHE program with that input, decrypts the ciphertext output, and compares it against a plaintext reference implementation that serves as ground truth. Inputs are sampled randomly within the valid domain using a fixed seed for reproducibility . Since results are aggregated over five independent runs with distinct seeds , the reported variance in Pass@1(*func*) reflects both LLM output stochasticity and sensitivity to input values.

## E  STRUCTURED PROMPT CREATION

Concretely, the composer agent's prompt is constructed by concatenating the fully verified C++ implementations of all required primitives — each already checked by the Security Verifier — into the Reference Pseudocode field of the standard formal prompt, followed by the high-level composition goal as the Objective. This supplies the agent with cryptographically sound building blocks directly in context, eliminating the need to regenerate or re-verify sub-routines during composition. We evaluate this strategy in detail in Section 5.3.

## F  RAG PIPELINE CREATION

The RAG knowledge base for TFHE is built from two primary sources: (i) the official TFHE C library headers and gate-bootstrapping API documentation, and (ii) expert-curated code snippets demonstrating correct API usage (e.g., proper bit-level encryption with `bootsSymEncrypt`, correct noise-aware gate sequencing). As illustrated in Figure 3b , human FHE experts perform a one-time offline annotation pass in which each function's docstring is enriched with a structured `@objective` tag encoding its cryptographic purpose, noise constraints, and usage preconditions, then paired with a representative code snippet. Retrieval operates over these annotated entries so that the agent receives snippets matched by cryptographic intent rather than surface-level keyword similarity. For CKKS, the corpus is constructed identically by substituting the TFHE documentation and annotation set with the corresponding CKKS library sources, confirming that RAG corpus construction is scheme-agnostic and requires no framework-level changes.

