# OpenReview forum: "FHE-Coder: Benchmarking Secure Agentic Code Generation for Fully Homomorphic Encryption"
_ICLR.cc/2026/Conference — ICLR 2026 Poster_

### Official Review · Reviewer_c3tK · 2025-10-31

**Soundness:** 3
**Presentation:** 3
**Contribution:** 3
**Rating:** 6
**Confidence:** 3

**Summary:**

The paper introduces TFHE-CODER, a three-stage agentic framework for secure Fully Homomorphic Encryption (FHE) programming. Unlike conventional code generation approaches that rely solely on post-hoc human inspection, TFHE-CODER enables large language models (LLMs) to proactively integrate security constraints during the generation process. The framework comprises three key components: FHE Prompt Formalizer, FHE API RAG Retriever, and FHE Security Verifier. To systematically evaluate the framework’s effectiveness, the authors constructed the TFHE-CODER Benchmark, consisting of ten FHE programming tasks. Experimental results demonstrate that large language models can autonomously generate FHE programs that are not only functionally correct and compilable but also verifiably secure.

**Strengths:**

1. The authors propose an end-to-end agentic framework explicitly designed around security. By integrating prompt formalization, retrieval, and automated security verification into a closed feedback loop, the system treats security correctness as a primary objective rather than a post-generation check.
2. The framework’s security verifier can automatically identify multiple classes of critical vulnerabilities and generate structured Formal Error Reports, enabling the agent to iteratively refine and correct insecure code.
3. The paper introduces a novel metric specifically tailored for security-critical code generation. This metric complements traditional functional correctness measures and prevents misleading conclusions based solely on task completion accuracy.

**Weaknesses:**

1. The RAG dictionary requires offline preparation and manual expert involvement, which limits the framework’s degree of automation and scalability.
2. The paper does not provide detailed measurements of computational overhead, runtime latency, or resource consumption introduced by the multi-stage verification process, making it difficult to assess practical efficiency.
3. While the framework performs well on simple FHE operations, its success rate drops significantly on more complex tasks.

**Questions:**

Please refer to my comments on weaknesses.

---

> ### Author Response · Authors · 2025-11-24
> **Response to reviewer c3tK**
>
> We sincerely thank Reviewer c3tK for their thoughtful review and valuable insights. We have carefully addressed the concerns raised in our detailed response below and have revised the manuscript to incorporate these key improvements.
>
> **Question 1. The RAG dictionary requires offline preparation and manual expert involvement, which limits the framework’s degree of automation and scalability.**
>
>
> The offline preparation of the RAG dictionary is a one-time, lightweight security step, not an ongoing manual effort. Its purpose is to eliminate API hallucinations—a common failure mode for baseline LLM agents in cryptographic domains—and to ensure that retrieval remains aligned with the strict semantics of the underlying HE library. This process is also fully scalable. Extending the framework to another scheme (e.g., CKKS) simply requires replacing the documentation corpus with that scheme’s library documentation. Likewise, scheme-specific keywords or security rules can be swapped in with minimal effort. The overall agentic workflow remains unchanged, making cross-scheme adaptation essentially plug-and-play. Thus, the small amount of offline preparation is both contained and necessary, and it enables the framework to generalize cleanly across different HE schemes and cryptographic tasks.We provide the results for the new HE scheme CKKS as follows.
>
> Table A: CKKS with GPT-5:
> | Workload     | Prior BAS p@1 (func.) | Prior BAS p@1 (sec.) | Prior COT p@1 (func.) | Prior COT p@1 (sec.) | Our FRS p@1 (func.) | Our FRS p@1 (sec.) |
> |-------------|------------------|----------------|------------------|----------------|------------------|----------------|
> | MatMul      | 1.00 ± 0.00      | 0.00 ± 0.00    | 0.80 ± 0.01      | 0.00 ± 0.00    | 0.80 ± 0.01      | 0.60 ± 0.02    |
> | Softmax     | 1.00 ± 0.00      | 0.00 ± 0.00    | 0.80 ± 0.03      | 0.00 ± 0.00    | 0.80 ± 0.02      | 0.60 ± 0.02    |
> | Attention   | 0.40 ± 0.02      | 0.00 ± 0.00    | 0.40 ± 0.01      | 0.00 ± 0.00    | 1.00 ± 0.00      | 1.00 ± 0.00    |
> | Transformer | 0.00 ± 0.00      | 0.00 ± 0.00    | 0.20 ± 0.01      | 0.00 ± 0.00    | 0.40 ± 0.01      | 0.60 ± 0.03    |

---

> ### Author Response · Authors · 2025-11-24
>
> **Question 2. The paper does not provide detailed measurements of computational overhead, runtime latency, or resource consumption introduced by the multi-stage verification process, making it difficult to assess practical efficiency.**
>
>
> We want to clarify that we provided the runtime latency analysis(compared to expert written ground truth implementation) in the paper (Figure 6, 7), and here we provide an explicit runtimes of the 3 selected workloads. The Compile Check simply executes a standard build command, and the critical Security Check is implemented as a lightweight static analysis of API usage, parameter configuration, and input encryption, meaning both complete in the order of milliseconds. Similarly, the Latency Check adds negligible overhead as it merely extracts execution metrics from the reference comparison. While the Functional Check is the most time-consuming step—ranging from seconds for simple gates like AND to minutes for complex circuits like Matrix-Matrix Multiplication—this duration is intrinsic to the execution of the FHE code itself rather than the framework's processing. Therefore, the computational overhead introduced specifically by our verification logic is minimal, as it relies primarily on highly efficient static analysis and standard system operations rather than heavy local computation.
>
> Note that there is no local computational overhead or resource consumption beyond actual runtime resources of testing the generation solutions in the verification module. The LLM models have been accessed via OpenRouter API so the only latency in the generation process is due to the API's own latency. We will make this explicit in the updated draft.
>
> | Task (Complexity) | Average Execution Time |
> | :--- | :--- |
> | **AND** (Elementary Gate Logic) | 10.45 milliseconds |
> | **ReLU** (Simple Bitwise Circuit) | 11.50 milliseconds |
> | **Matrix Multiplication** (Complex Compositional Circuit) | 2.54 minutes |

---

> ### Author Response · Authors · 2025-11-24
>
> **Question 3. While the framework performs well on simple FHE operations, its success rate drops significantly on more complex tasks.**
>
> We wish to clarify that the performance drop is not significant when using the proposed structured decomposition technique. The results shown in Figure 6 correspond to an intermediate generation step before applying our structured decomposition, included only to illustrate why unstructured generation struggles with compositional complexity. They do not reflect the full capability of the framework. When the complete method is applied—specifically the Structured Decomposition strategy from Section 5.3, where the agent first generates and verifies smaller primitives before composing them—the success rate remains robust even on complex tasks. As shown in Table 2, this restores the security pass rate to 1.0 for matrix operations and 0.8 for CNNs. Importantly, this scalability is not limited to TFHE. Our new results on CKKS, a fundamentally different HE scheme, show the same trend: baseline agents collapse on complex tasks, while the structured prompting in our framework consistently recovers both functional and secure implementations.
>
> Also, we extend our approach to even more advanced architectures—Attention and Transformer modules (Table A in the responses of Question 1) —demonstrating that the framework maintains high reliability across state-of-the-art nonlinear circuits and multiple HE schemes.

---

### Official Review · Reviewer_er3J · 2025-11-01

**Soundness:** 2
**Presentation:** 3
**Contribution:** 2
**Rating:** 4
**Confidence:** 4

**Summary:**

This paper targets an important problem of automatically generating secure API calls and parameters for Fully Homomorphic Encryption over Torus (TFHE). It proposes to leverage the power of agentic large language models (LLMs) to generate TFHE API calls and parameters, with the assistance of Retrieval-Augmented Generation (RAG) and post validation techniques. Evaluation on a set of tasks shows that the method outperforms vanilla prompting and Chain-of-Thought (CoT) prompting.

**Strengths:**

- Addresses an important and practical problem of generating secure TFHE API calls and parameters.
- Clear writing and well-structured presentation.

**Weaknesses:**

- The evaluation tasks are limited in complexity, diversity, and reproducibility.
- No guarantee on the security and functional correctness of the generated programs.
- Limited novelty in the technical contributions, where RAG and post validation are not new.

**Questions:**

Generating secure API calls and parameters for TFHE is an important problem, as developers often lack cryptographic expertise in programming TFHE applications. The proposed method leverages agentic LLMs with enhancements like RAG and post validation to improve the generation accuracy. However, there are several concerns over the empirical evaluation and technical contributions.

The evaluation of the proposed method is limited. The benchmark tasks are simple Machine Learning (ML) tasks, which are as simple as vector/matrix multiplications and basic ML models like MLP or CNN. These tasks are often too simple to demonstrate the effectiveness of the proposed method, as human developers can also easily implement these tasks. It would be more convincing to show the effectiveness of the method on more complex and diverse tasks, including natural language processing models, transformer-based models, and other non-ML tasks. Additionally, the reproducibility of the evaluation is not well demonstrated. Although the paper mentions that the seeds are fixed, the randomness in LLM generation can still lead to variance in results. It would be helpful to repeat each experiment multiple times and report the average performance with confidence intervals.

There is limited guarantee on the security and functional correctness of the generated TFHE programs. While the post validation step is a safe-guard to filter out incorrect programs, the paper cannot ensure that the validation conditions are correct and comprehensive. For instance, how do we know that the verification conditions in Dafny code are sufficient to eliminate all security vulnerabilities? Also, it is not clear that how the proposed method can ensure that the generated conditions are aligned with the developer's intent. More discussion on these aspects would be helpful.

The technical contributions of the paper appear incremental. The use of RAG and post validation are not new techniques in the context of LLMs. This work is a simple application of such techniques to the TFHE domain. The contributions would be stronger if such a technique can also extend to other cryptographic domains other than TFHE, such as MPC, partial HE, or ZK proofs.

---

> ### Author Response · Authors · 2025-11-24
> **Response to reviewer er3J**
>
> We sincerely thank Reviewer er3J for their insightful feedback and the time dedicated to reviewing our work. We have carefully addressed these points in our detailed response below and have incorporated the corresponding revisions into the updated manuscript.
>
> **Question 1. The evaluation of the proposed method is limited. The benchmark tasks are simple Machine Learning (ML) tasks, which are as simple as vector/matrix multiplications and basic ML models like MLP or CNN. These tasks are often too simple to demonstrate the effectiveness of the proposed method, as human developers can also easily implement these tasks. It would be more convincing to show the effectiveness of the method on more complex and diverse tasks, including natural language processing models, transformer-based models, and other non-ML tasks.**
>
>
> While the benchmark tasks (MLP or CNN) appear simple in the plaintext domain, they become substantially more difficult under homomorphic encryption (HE). Even implementing basic arithmetic or simple ML models in HE requires navigating a sequence of tightly coupled steps:
> (1) encoding high-dimensional data into plaintext SIMD slots,
> (2) encrypting with scheme-consistent parameters and noise budgets,
> (3) executing ciphertext operations while tracking noise growth and modulus consumption, and
> (4) decrypting and decoding while preserving correctness and security.
>
> Each step interacts with cryptographic parameters in subtle ways, and small mistakes—such as mismanaging noise, choosing inconsistent parameters, or using an API incorrectly—lead to silent decryption failures or security violations. Prior work[f, g, h] has repeatedly shown that even “simple” ML workloads become scientifically challenging when ported to FHE due to these parameterized, noise-sensitive computation paths. **This is why community competitions such as FHERMA in 2025 still focus on tasks as basic as CNN[f] and MLP[g] inference: these workloads are already non-trivial in an encrypted setting and routinely fail in baseline LLM agents.**
>
> Our method is effective precisely because it addresses these intrinsic scientific challenges—formalizing parameters, guiding correct API sequences, and enforcing noise-safe operator construction.
>
> **To further demonstrate generality, as Table C shows, we also evaluate our framework on more workloads such as Softmax, Attention blocks, and Transformer components, which involve deeper circuits and more complex noise dynamics**. Our system (FRS) maintains functional correctness while significantly outperforming baselines (BAS and COT) on security, demonstrating that it scales to more complex and diverse encrypted workloads.
>
> Table C: TFHE with GPT-5:
> | Workload     | BAS p@1 (func.) | BAS p@1 (sec.) | COT p@1 (func.) | COT p@1 (sec.) | FRS p@1 (func.) | FRS p@1 (sec.) |
> |-------------|------------------|----------------|------------------|----------------|------------------|----------------|
> | MatMul      | 0.00 ± 0.00      | 0.00 ± 0.00    | 0.00 ± 0.00      | 0.00 ± 0.00    | 0.80 ± 0.05      | 1.00 ± 0.00    |
> | Softmax     | 0.80 ± 0.05      | 0.00 ± 0.00    | 0.60 ± 0.02      | 0.00 ± 0.00    | 1.00 ± 0.00      | 1.00 ± 0.00    |
> | Attention   | 0.00 ± 0.00      | 0.00 ± 0.00    | 0.20 ± 0.01      | 0.00 ± 0.00    | 0.80 ± 0.02      | 0.80 ± 0.02    |
> | Transformer | 0.00 ± 0.00      | 0.00 ± 0.00    | 0.20 ± 0.01      | 0.00 ± 0.00    | 0.40 ± 0.01      | 0.80 ± 0.02 |
>
> [a] https://machinelearning.apple.com/research/homomorphic-encryption
>
> [b] https://docs.zama.org/concrete-ml
>
> [c] https://docs.z1labs.ai/cyphers-fhevm-technology/what-is-fhevm
>
> [d] https://www.finopotamus.com/post/zama-raises-57m-in-series-b-to-bring-end-to-end-encryption-to-public-blockchains
>
> [e] https://www.microsoft.com/en-us/research/blog/password-monitor-safeguarding-passwords-in-microsoft-edge/
>
> [f] https://fherma.io/challenges/652bf663485c878710fd0209/overview
>
> [g] https://fherma.io/challenges/652bf648485c878710fd0208/overview
>
> [h] Dathathri, Roshan, et al. "CHET: an optimizing compiler for fully-homomorphic neural-network inferencing." Proceedings of the 40th ACM SIGPLAN conference on programming language design and implementation.
>
> [i] Albrecht, Martin R., Rachel Player, and Sam Scott. "On the concrete hardness of learning with errors." Cryptology ePrint Archive (2015).

---

> ### Author Response · Authors · 2025-11-24
>
> **Question 2. Additionally, the reproducibility of the evaluation is not well demonstrated. Although the paper mentions that the seeds are fixed, the randomness in LLM generation can still lead to variance in results. It would be helpful to repeat each experiment multiple times and report the average performance with confidence intervals.**
>
>
> We clarify that our evaluation already accounts for the inherent randomness of LLM generation. As stated in Section 4.3 (lines 296–298), all reported numerical results—including Pass@1 for functionality and security, as well as latency—are computed as the average of five independent runs for every task and baseline. This five-run averaging, combined with fixed inference seeds, provides stable and reproducible metrics. In the revised manuscript, we will additionally report confidence intervals to make variance explicit, and we will open-source our code to ensure full reproducibility of all experiments.

---

> ### Author Response · Authors · 2025-11-24
>
> **Question 3.There is limited guarantee on the security and functional correctness of the generated TFHE programs. For instance, how do we know that the verification conditions in Dafny code are sufficient to eliminate all security vulnerabilities?**
>
>
>
> We want to clarify that the security guarantees of our generated code are fundamentally rooted in the mathematical hardness of the Learning With Errors (LWE) problem[i], which ensures that as long as the accumulated noise remains below a specific decryption threshold, the cryptographic operations are secure. To enforce this theoretical guarantee in practice, our framework integrates two formal layers: the LWE Estimator[i] and Dafny specifications. The Lattice Estimator mathematically solves for the precise security parameters ($\lambda$) to constrain the LLM within rigorous cryptographic bounds. Complementing this, the Dafny pseudocode serves as a critical mechanism to eliminate logic-based vulnerabilities by translating high-level intent into formal specifications with ensure statements. These specifications guide the agent to embed assert checks directly into the C code, ensuring that the generated solution maintains structural correctness and satisfies necessary invariants at every intermediate step of the homomorphic circuit.
>
> Regarding functional correctness, we utilize standard unit testing to verify that the generated logic aligns with the developer's intent; a functionally correct code will consistently pass these tests (high Pass@k(func)), while faulty logic will fail. Standard testing alone is insufficient because, as our results show, our baseline agents often produce code that is functionally correct (passes unit tests) but cryptographically insecure (uses plaintext). Our proposed security verification layer bridges this gap, ensuring that code is not only functional but also adheres to the strict privacy constraints of the TFHE scheme.

---

> ### Author Response · Authors · 2025-11-24
>
> **Question 4. Also, it is not clear that how the proposed method can ensure that the generated conditions are aligned with the developer's intent. More discussion on these aspects would be helpful.**
>
> We want to clarify that the cryptographic security of the generated code is guaranteed by the mathematical hardness of the Learning With Errors (LWE) problem, which is enforced in our framework by integrating the LWE Security tool [i] to mathematically solve for the precise security parameters ($\lambda$) required for the task.
>
> However, ensuring alignment with the developer's functional intent is primarily the role of our Functional Check. We utilize rigorous unit testing as the definitive metric for intent alignment; a generated program that correctly implements the user's intended logic will consistently pass these tests (reflected in a high Pass@k(func) score), while any deviation from the intended logic results in failure. By requiring the generated solution to successfully pass both this Functional Check (validating intent) and the Security Check (validating LWE compliance via parameter enforcement), our framework uniquely certifies that the final code is simultaneously functionally accurate and cryptographically secure.

---

> ### Author Response · Authors · 2025-11-24
>
> **Question 5. The technical contributions of the paper appear incremental. The use of RAG and post validation are not new techniques in the context of LLMs. This work is a simple application of such techniques to the TFHE domain. The contributions would be stronger if such a technique can also extend to other cryptographic domains other than TFHE, such as MPC, partial HE, or ZK proofs.**
>
>
> Although our framework uses components such as RAG and validation, applying these techniques to TFHE is not incremental. TFHE programming has strict cryptographic semantics, parameter constraints, and noise rules that cause vanilla prompting, CoT, and standard RAG to fail almost entirely, as shown by their near-zero security pass rates (Table B). The difficulty arises because LLMs lack the mathematical structure needed to select secure parameters, respect ciphertext-only computation, or interpret cryptographic APIs.
>
> Our contributions lie in three required adaptations that make LLM-based FHE programming possible:
> (1) a Prompt Formalizer that mathematically derives secure parameters and valid ciphertext structures;
> (2) an API RAG Retriever built with expert-enriched metadata to bridge the semantic gap between natural-language intent and cryptographic APIs; and
> (3) a Security Verifier that enforces core cryptographic invariants that functional tests alone cannot capture.
>
> These components form a general recipe for cryptographic domains. Extending to MPC or ZKP only requires substituting domain-specific rules and security invariants in these modules.
>
> To demonstrate generality, we evaluated CKKS—an FHE scheme fundamentally different from TFHE—and observed the same pattern: baseline agents fail on both correctness and security, while our framework consistently succeeds (Table A).
>
>
> Table A: CKKS with GPT-5:
> | Workload     | Prior BAS p@1 (func.) | Prior BAS p@1 (sec.) | Prior COT p@1 (func.) | Prior COT p@1 (sec.) | Our FRS p@1 (func.) | Our FRS p@1 (sec.) |
> |-------------|------------------|----------------|------------------|----------------|------------------|----------------|
> | MatMul      | 1.00 ± 0.00      | 0.00 ± 0.00    | 0.80 ± 0.01      | 0.00 ± 0.00    | 0.80 ± 0.01      | 0.60 ± 0.02    |
> | Softmax     | 1.00 ± 0.00      | 0.00 ± 0.00    | 0.80 ± 0.03      | 0.00 ± 0.00    | 0.80 ± 0.02      | 0.60 ± 0.02    |
> | Attention   | 0.40 ± 0.02      | 0.00 ± 0.00    | 0.40 ± 0.01      | 0.00 ± 0.00    | 1.00 ± 0.00      | 1.00 ± 0.00    |
> | Transformer | 0.00 ± 0.00      | 0.00 ± 0.00    | 0.20 ± 0.01      | 0.00 ± 0.00    | 0.40 ± 0.01      | 0.60 ± 0.03    |
>
> Table B: Results for Using Vanilla RAG with Documentation (without preprocessing with expert summary):
>
> | Workload | pass@1 (func.) | pass@1 (sec.) |
> |----------|----------------|---------------|
> | ReLU     | 0.40 ± 0.02    | 0.20 ± 0.01   |
> | MatMul   | 0.20 ± 0.01    | 0.00 ± 0.00   |

---

### Official Review · Reviewer_jFYL · 2025-11-01

**Soundness:** 3
**Presentation:** 3
**Contribution:** 3
**Rating:** 6
**Confidence:** 3

**Summary:**

This paper investigates the potential of Large Language Model (LLM) agents to automate the generation of secure Fully Homomorphic Encryption over the Torus (TFHE) code from natural language. (Fig 2 of the paper gives a giid outline of the method.) The role of LLMs is to respond to user intent, configure secure parameters,  adapt the API, and leverage an automated security verifier that provides iterative feedback to correct cryptographic flaws. The system produces code that is compilable, functionally correct, and verifiably secure.  The focus on TFHE, as opposed to other FHE schemes, is based on the  efficient gate bootstrapping and functional bootstrapping, which allow computation of arbitrary functions while refreshing noise.

**Strengths:**

This seems to be a good case study of using LLMs, in combination with structured workflow and other tools, to solve a specific programming problem with modest scope.  The focus on TFHE will be a plus for specific audience.

**Weaknesses:**

From an application perspective, the focus on TFHE is specialized, limiting the apparent audience for this work.  How many people need TFHE code and how much variety is there among possible users?
From a scientific perspective, it is not clear how much of the difficulty, or how much of the apparent success of the approach is due to particular characteristics of TFHE.
Further, it is hard to see from the conference-length writeup how broad the solution is (how different are different requests for TFHE code?) and how compelling the verification is.   A little more information on how fully the code is verified would be helpfui.

**Questions:**

What would be needed to generalize this work beyond TFHE?  There are likely other programming tasks where LLMs could have relevant knowledge about parameters and there could be appropriate verification techniques.  If someone provided versions of FHE Prompt Formalizer, FHE API RAG Retriever and FHE Security Verifier. for another problem, would hte method be likely to work?  Why or why not?

---

> ### Author Response · Authors · 2025-11-24
> **Response to reviewer jFYL**
>
> We appreciate Reviewer jFYL’s constructive feedback and the time spent reviewing our paper. We have provided detailed responses and incorporated the key revisions into the manuscript.
>
> **Question 1: From an application perspective, the focus on TFHE is specialized, limiting the apparent audience for this work. How many people need TFHE code and how much variety is there among possible users?**
>
>
> TFHE is one of the most widely deployed and industry-relevant Homomorphic Encryption (HE) schemes, with adoption that spans billions of devices and multiple major technology platforms. Apple, for instance, has integrated HE-based private contact discovery in iMessage and private email retrieval since iOS 18 (2024)[a], directly benefiting more than 1.5 billion iPhone users as of 2025. Other large-scale deployments of HE include Microsoft’s Browser Password Monitor [e]. More specifically, TFHE has become a cornerstone of modern encrypted computation stacks: Zama’s TFHE-based ecosystem supports encrypted machine learning (Concrete ML[b]), private smart contracts (fhEVM[c]), and blockchain infrastructures[d]. More TFHE programming will be needed for emerging applications.
>
> Equally important, TFHE’s development workflow—parameter selection, API-usage correctness, noise budgeting, and secure circuit composition—is highly representative of workflows in other major HE schemes such as CKKS. As a result, our framework is not limited to TFHE: the three components (prompt formalization, RAG-based API retrieval, and security verification) extend naturally to all HE schemes by swapping the parameter estimator and API corpus, as shown in the repsonses for Question 4.
>
> [a] https://machinelearning.apple.com/research/homomorphic-encryption
>
> [b] https://docs.zama.org/concrete-ml
>
> [c] https://docs.z1labs.ai/cyphers-fhevm-technology/what-is-fhevm
>
> [d] https://www.finopotamus.com/post/zama-raises-57m-in-series-b-to-bring-end-to-end-encryption-to-public-blockchains
>
> [e] https://www.microsoft.com/en-us/research/blog/password-monitor-safeguarding-passwords-in-microsoft-edge/
>
> [f] https://fherma.io/challenges/652bf663485c878710fd0209/overview
>
> [g] https://fherma.io/challenges/652bf648485c878710fd0208/overview
>
> [h] Dathathri, Roshan, et al. "CHET: an optimizing compiler for fully-homomorphic neural-network inferencing." Proceedings of the 40th ACM SIGPLAN conference on programming language design and implementation.
>
> [i] Albrecht, Martin R., Rachel Player, and Sam Scott. "On the concrete hardness of learning with errors." Cryptology ePrint Archive (2015).

---

> ### Author Response · Authors · 2025-11-24
>
> **Question 2. From a scientific perspective, it is not clear how much of the difficulty, or how much of the apparent success of the approach is due to particular characteristics of TFHE.**
>
> TFHE coding is inherently difficult due to the fundamental characteristics of homomorphic encryption. Implementing even basic HE functionality requires navigating a sequence of tightly coupled and error-prone steps:
> (1) encoding high-dimensional messages into plaintext SIMD structures,
> (2) encrypting them under parameters that dictate the available noise budget,
> (3) performing ciphertext operations while monitoring noise growth and modulus consumption, and
> (4) decrypting and decoding while ensuring both correctness and security.
>
> Prior work[f, g, h] has repeatedly demonstrated that hand-written HE implementations are extremely challenging, precisely because each step interacts with cryptographic parameters in subtle and non-obvious ways. Small mistakes—such as mismanaging noise, selecting inconsistent parameters, or invoking APIs incorrectly—can lead to silent security failures or incorrect outputs.
>
> This difficulty is further illustrated by recent community competition benchmarks such as the 2025 FHERMA competition, where tasks as basic as CNN inference[f] or MLP inference[g] must be manually implemented in HE libraries. The fact that such standard models still require dedicated competitions underscores that the complexity is intrinsic and widely recognized.

---

> ### Author Response · Authors · 2025-11-24
>
> **Question 3. How compelling the verification is. A little more information on how fully the code is verified would be helpful.**
>
>
> Our verification is compelling because it provides complete coverage of the two essential dimensions of secure FHE code: cryptographic security and functional correctness. For security, our verifier is grounded in the mathematical hardness of learning with errors (LWE)[i], not heuristics. It checks that all parameters satisfy the LWE Estimator’s security bounds and confirms that only homomorphic APIs are used, preventing any unintended plaintext leakage[i]. For functionality, we validate that the generated code matches the developer’s intended computation through comprehensive unit tests and correctness checks. By requiring a solution to simultaneously pass both the LWE-based security constraints and the logic-based functional tests, our framework ensures the code is fully verified—both mathematically secure and operationally correct.

---

> ### Author Response · Authors · 2025-11-24
>
> **Question 4. How broad the solution is and what would be needed to generalize this work beyond TFHE and other tasks?**
>
> Although our experiments use TFHE as a representative case study, the framework is designed to generalize broadly to other HE schemes and other specialized programming tasks with minimal effort.
>
> (1) Generalization to other FHE schemes.
> All three modules in our system require only lightweight substitutions:
> – the Prompt Formalizer loads the target scheme’s parameter rules (i.e., key words replacement),
> – the API RAG Retriever ingests the target library’s documentation (i.e., existing documatation replacement), and
> – the Security Verifier swaps in the scheme’s own security invariants (i.e., 3 security parameters replacement).
> The agentic workflow itself remains unchanged, making cross-scheme adaptation essentially plug-and-play.
>
> (2) Generalization to other programming domains.
> The framework follows a domain-agnostic recipe—formalize constraints, retrieve correct APIs, and verify correctness. Any domain that provides constraint definitions, documentation, and testable verification criteria can be supported without modifying the agentic loop. This modular, feedback-driven design naturally extends to tasks beyond HE.
>
> To demonstrate this breadth, we include experiments on CKKS (an FHE scheme fundamentally different from TFHE) and on more complex workloads as the Table A shows. Direct LLM agents (BAS, CoT) fail to consistently produce secure and correct code, while our method (FRS) reliably satisfies both requirements. We also evaluate additional TFHE tasks, as Table B shows, such as Attention and Transformer circuits, all showing the same trend, further confirming strong generalization beyond the initial TFHE examples.
>
>
> Table A: CKKS with GPT-5:
> | Workload     | Prior BAS p@1 (func.) | Prior BAS p@1 (sec.) | Prior COT p@1 (func.) | Prior COT p@1 (sec.) | Our FRS p@1 (func.) | Our FRS p@1 (sec.) |
> |-------------|------------------|----------------|------------------|----------------|------------------|----------------|
> | MatMul      | 1.00 ± 0.00      | 0.00 ± 0.00    | 0.80 ± 0.01      | 0.00 ± 0.00    | 0.80 ± 0.01      | 0.60 ± 0.02    |
> | Softmax     | 1.00 ± 0.00      | 0.00 ± 0.00    | 0.80 ± 0.03      | 0.00 ± 0.00    | 0.80 ± 0.02      | 0.60 ± 0.02    |
> | Attention   | 0.40 ± 0.02      | 0.00 ± 0.00    | 0.40 ± 0.01      | 0.00 ± 0.00    | 1.00 ± 0.00      | 1.00 ± 0.00    |
> | Transformer | 0.00 ± 0.00      | 0.00 ± 0.00    | 0.20 ± 0.01      | 0.00 ± 0.00    | 0.40 ± 0.01      | 0.60 ± 0.03    |
>
>
> Table B: TFHE with GPT-5:
>
> | Workload     | BAS p@1 (func.) | BAS p@1 (sec.) | COT p@1 (func.) | COT p@1 (sec.) | FRS p@1 (func.) | FRS p@1 (sec.) |
> |-------------|------------------|----------------|------------------|----------------|------------------|----------------|
> | MatMul      | 0.00 ± 0.00      | 0.00 ± 0.00    | 0.00 ± 0.00      | 0.00 ± 0.00    | 0.80 ± 0.05      | 1.00 ± 0.00    |
> | Softmax     | 0.80 ± 0.05      | 0.00 ± 0.00    | 0.60 ± 0.02      | 0.00 ± 0.00    | 1.00 ± 0.00      | 1.00 ± 0.00    |
> | Attention   | 0.00 ± 0.00      | 0.00 ± 0.00    | 0.20 ± 0.01      | 0.00 ± 0.00    | 0.80 ± 0.02      | 0.80 ± 0.02    |
> | Transformer | 0.00 ± 0.00      | 0.00 ± 0.00    | 0.20 ± 0.01      | 0.00 ± 0.00    | 0.40 ± 0.01      | 0.80 ± 0.02    |

---

### Comment · Area_Chair_wz4p · 2025-11-24

Dear Authors and Reviewers,

I would like to encourage the  reviewers to carefully read not only the authors rebuttals but also the other reviews to start engaging in a discussion.

Best regards
AC

---

### Author Response · Authors · 2025-11-29
**Thanking all the reviewers and summary of concerns and resolution for AC**

We thank the reviewers for their constructive feedback. We have uploaded a revised manuscript and detailed responses. Below is a summary of the common questions raised across reviews and our new experimental results demonstrating robustness and generalization.

------------------

**We appreciate the reviewers' recognition of our work along the following dimensions:**
* **Timeliness & Importance:** The paper targets a critical skills gap in the specialized but high-impact domain of TFHE applications **(Reviewer jFYL )**.
* **Novel Framework:** The proposed three-phase agentic workflow is recognized as a structured solution to the complexity of cryptographic code generation **(Reviewer jFYL, Reviewer c3tK )**.
* **Methodological Clarity:** The decomposition of the problem into formalization, retrieval, and verification is acknowledged as a logical approach to democratizing secure computation **(Reviewer c3tK )**.

------------------
**Regarding the reviewers’ concerns, we have carefully addressed all raised issues in our detailed responses and revised the manuscript accordingly. For easy navigation, we indicate the corresponding locations in our responses (Reviewer ID + Question Number) in parentheses.**

* **Industrial Relevance and Broad Adoption:** We addressed concerns regarding the "specialized" nature of our focus by highlighting TFHE's massive industrial footprint. We detailed its deployment in **Apple’s iOS 18** (protecting 1.5 billion users) , **Microsoft’s password monitoring** , and **Zama’s** blockchain ecosystems. We clarified that TFHE is a cornerstone of modern privacy infrastructure, not just an academic niche **(see Reviewer jFYL Q1)**.

* **Generalization to other schemes (CKKS) and complex architectures:** We clarified that our framework is a generalizable blueprint, not limited to TFHE. We added new experiments adapting the framework to the **CKKS** scheme (a fundamentally different arithmetic scheme) with minimal configuration changes. Furthermore, we introduced new results on complex non-linear workloads, including **Attention mechanisms and Transformers**, demonstrating that our Structured Decomposition strategy enables the synthesis of state-of-the-art architectures where baselines fail **(see Reviewer jFYL Q4 , Reviewer er3J Q1 , Reviewer c3tK Q3 )**.

* **Rigor of security verification and user intent:** We clarified that our security guarantees are not heuristic but rooted in the mathematical hardness of the **Learning With Errors (LWE)** problem via the **Lattice/LWE Estimator**. We explained how functional intent is guaranteed through rigorous unit testing, while security is enforced by parameter constraints. This dual-verification approach ensures code is both operationally correct and mathematically secure **(see Reviewer jFYL Q3 , Reviewer er3J Q3–Q4 )**.

* **Novelty of RAG and Validation adaptations:** We addressed concerns about incrementalism by demonstrating that "vanilla" RAG and standard validation fail almost entirely in the FHE domain (near-zero security pass rates). We detailed our three specific novelties: LWE-grounded Prompt Formalization, Expert-Enriched RAG to bridge the semantic gap, and a Security Verifier that enforces cryptographic invariants standard tests miss **(see Reviewer er3J Q5 )**.

* **Scalability and automation of RAG:** We clarified that the offline preparation for the RAG retriever is a one-time security investment to prevent API hallucinations. Additionally, we outlined the path to full automation using an LLM-based Formalizer Agent and emphasized that extending to new schemes is a plug-and-play process of swapping documentation corpora **(see Reviewer c3tK Q1 )**. We show example extension to CKKS scheme.

* **Computational overhead and latency:** We provided a granular breakdown of execution times, clarifying that the verification overhead (milliseconds for static analysis) is negligible compared to the intrinsic runtime of FHE execution. We also clarified that generation latency is dominated by external API calls, not local compute **(see Reviewer c3tK Q2 )**.

* **Reproducibility:** We confirmed that all reported metrics are 5-run averages with fixed seeds to mitigate LLM randomness. We have included confidence intervals in the revised manuscript and commit to open-sourcing our code **(see Reviewer er3J Q2 )**.
------------------
We believe this work takes a significant step toward democratizing secure computation by lowering the expertise barrier for privacy-preserving machine learning. We once again thank the reviewers, the Area Chair, and the Program Committee for their time and thoughtful evaluation.

---

### Meta-Review · Area_Chair_57aA · 2026-01-05

**Summary:**

The rebuttal substantially improves clarity and scope. However, claims about full verification, elimination of vulnerabilities and generality should be carefully revised to avoid overstating guarantees.

**Reviewer Concerns:**

Strength of security and correctness guarantees (raised by two reviewers): verification scope still looks overclaimed

Generality beyond FHE: argued conceptually but not demonstrated.

Most of the other concerns seem to be addressed

**Reviewer Scores:**

I believe reviewers keep their original scores. However, it's important to note that two reviewers are fairly confident (one with score 6 even doesn't understand the title of the paper and no expertise in this area). The AC is also not familiar with this topic and no expertise in this area. This makes the final decision quite challenging

---

### Decision · Program_Chairs · 2026-01-26

Accept (Poster)